# SUBJECT-INVARIANT DOMAIN GENERALIZATION FOR PSYCHIATRIC DISORDER IDENTIFICATION

## ABSTRACT

Analyzing functional brain networks has emerged as a critical approach for understanding and diagnosing psychiatric disorders. Existing approaches primarily follow the standard supervised learning, which assumes that source and target data are independent and identically distributed. However, due to substantial inter-subject distributional differences in brain network data, models built on this assumption struggle to generalize from source to target datasets, resulting in suboptimal diagnostic performance. To address this issue, we propose a two-stage Subject-Invariant Domain Generalization (SIDG) model that learns subject-invariant representations in the pre-training stage, enabling their effective use for better psychiatric disorder identifcation in the fine-tuning stage. In order to overcome the mismatch between single-level topological representation methods and the inherently hierarchical topology of brain networks, we introduce a novel Hierarchical Topology Enhanced Graph Transformer Reconstruction (HTE-GTR) module to thoroughly learn subject-invariant representations distributed across multiple topological levels. Furthermore, we design tailored Subject-Invariant Reconstruction (SIR) loss comprising a subject-invariant term and a reconstruction term, to mitigate the impact of inter-subject distributional differences while preserving discriminative information for downstream tasks. Experiment results show clear improvements of our proposed SIDG on both the public ABIDE and ADHD datasets. The code is available at https://anonymous.4open.science/r/SIDG.

## 1 INTRODUCTION

Brain psychiatric disorders such as autism spectrum disorder (ASD) and attention deficit hyperactivity disorder (ADHD) impact the quality of life of hundreds of millions globally (Lord et al., 2020; Da Silva et al., 2023). These disorders involve complex neurobehavioral and neurobiological features, making accurate diagnosis particularly challenging (Andreazza et al., 2025). In recent years, functional magnetic resonance imaging (fMRI) has shown remarkable promise for both research and clinical applications (Biswal & Uddin, 2025). By capturing blood-oxygen-level-dependent (BOLD) signals, fMRI enables the assessment of functional connectivity (FC) among brain regions of interest (ROIs) (Li et al., 2025). Analyzing these connectivity patterns can reveal abnormal brain networks, providing insights for diagnosis and treatment (Peng et al., 2025).

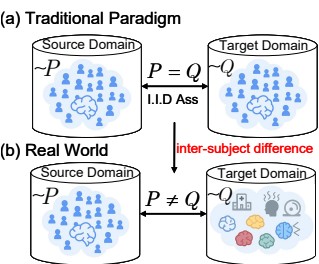

Figure 1: An illustration comparing the traditional identification paradigm with real-world scenarios.

Existing methods for classifying psychiatric disorders mainly follow standard supervised learning (Peng et al., 2025; Pei et al., 2025). Several approaches based on Graph Neural Network (GNN) and Graph Transformer (GT) have shown promising results under this paradigm. (Peng et al., 2024a; 2025). In these approaches, models are trained to directly classify each subject as a patient or healthy control (HC), using pooled subject-level features and labels from all available data. A fundamental assumption (Fig. 1) of this paradigm is that training (source) and testing (target) data are independent and identically distributed (I.I.D) (Cunningham et al., 2008). However, there are substantial inter-subject distributional differences in brain network data, which

cause models built on this assumption to generalize poorly from source to target datasets, resulting in suboptimal diagnostic performance.

Facing this bottleneck, we find that Domain Generalization (DG) methods (Zhou et al., 2022) provide a tailored solution through generating domain-invariant representations. Our insight is to design a DG method that learns subject-invariant representations, thereby addressing the inherent weakness of supervised learning models in handling inter-subject distributional differences. To this end, two fundamental challenges must be addressed. On the one hand, existing DG approaches focus on constructing group-invariant models, such as sex-invariant (Peng et al., 2024a), health-status-invariant (Kang et al., 2023), and site-invariant representations (Yu et al., 2025), but they lack explicit subject-invariant modeling, making it difficult to fully mitigate inter-subject variability. On the other hand, the inherently hierarchical topology of brain networks requires graph representation methods capable of handling multi-level information. (Park & Friston, 2013; Yeh, 2022; Hilgetag & Goulas, 2020). Yet, most current methods focus on a single topological level (Yu et al., 2025; Mao et al., 2024), limiting their ability to extract subject-invariant patterns across multiple levels.

To tackle the aforementioned challenges, we propose a two-stage Subject-Invariant Domain Generalization (SIDG) model that first learns subject-invariant representations in pre-training and then leverages them for effective psychiatric disorder classification in the fine-tuning stage. Specifically, we introduce the Hierarchical Topology Enhanced Graph Transformer Reconstruction (HTE-GTR), which constructs a hierarchical graph with distinct topological views and designs level-specific attention mechanisms to capture subject-invariant patterns across multiple levels, thereby effectively leveraging the intrinsic hierarchical structure of brain networks. Moreover, to guide model mitigate the impact of inter-subject distributional differences while preserving discriminative information for downstream tasks, we design the Subject-Invariant Reconstruction (SIR) loss, comprising a subject-invariant term and a reconstruction term. The subject-invariant term enforces intra-subject consistency while removing inter-subject variations, and the reconstruction term preserves the discriminative quality of the representations for downstream tasks.

The main contributions of this paper are summarized as follows:

(1) We propose a novel SIDG model with a pre-training–fine-tuning paradigm that effectively learns subject-invariant representations, substantially mitigating the impact of substantial inter-subject distributional differences for the first time.

(2) We introduce the HTE-GTR module, specifically designed to capture subject-invariant patterns distributed across multiple topological levels, aligning with the inherently hierarchical topology of brain networks.

(3) We designed a tailored SIR loss, consisting of subject-invariant term and a reconstruction term, to both mitigate the impact of inter-subject distributional differences and preserve discriminative information for downstream classification.

## 2 RELATED WORK

**Graph Supervised Learning.** Existing supervised graph learning methods for brain disorder identification fall into GNN- and GT-based approaches. In particular, GNN-based methods have been developed with diverse methodological strategies. For instance, AGE-GCN was proposed to enhance dissimilarities of brain regions (Ding et al., 2025), while GroupBNA was designed to adapt to distinct subject groups and improve robustness (Peng et al., 2024a). The DSVB framework focuses on modeling time-varying topological structures (Yap et al., 2024). Meanwhile, BrainHGL (Wen et al., 2025), STW-HCN (Liu et al., 2024b), and HSGNN (Chen et al., 2025) were proposed to capture more complex or heterogeneous connectivity patterns. CRGNN (Xia et al., 2024) and BrainIB (Zheng et al., 2025) were proposed to enhance task adaptability and informative subgraph selection. In addition, LG-GNN (Zhang et al., 2023) incorporates both non-imaging subject information and inter-subject relationships to pinpoint disease-related regions and biomarkers, while MAHGCN (Liu et al., 2024c) builds on stacked graph convolutional layers with adaptive pooling for comprehensive extraction of diagnostic knowledge. Beyond GNN-based methods, GT-based approaches have also been developed for brain disorder identification (Kan et al., 2022). BioBGT encodes the small-world architecture of brain graphs (Peng et al., 2025). CAGT (Pei et al., 2025) and Com-BrainTF (Bannadabhavi et al., 2023) integrate community information of subnetworks

and topological properties into transformer architectures. ALTER leverages biased random walks to capture long-range dependencies among ROIs (Yu et al., 2024), while KAGT incorporates a domain adaptation module to alleviate data heterogeneity (Song et al., 2025). Gradformer emphasizes structural inductive biases critical for graph tasks (Liu et al., 2024a). Contrasformer constructs a prior-knowledge-enhanced contrast graph with a two-stream attention mechanism to address distribution shifts across sub-populations (Xu et al., 2024). GBT employs an AWMA-based transformer module and a geometric-oriented representation learning module for fMRI connectome analysis (Peng et al., 2024b). Although graph supervised learning methods have shown promise in classifying psychiatric disorders, substantial distributional differences across subjects pose challenges for models trained on source data to generalize to target data, thereby limiting diagnostic accuracy.

**Group-Invariant Model.** Existing DG methods for brain disorder diagnosis mainly focus on constructing group-invariant models. GroupBNA (Peng et al., 2024a), EAG-RS (Jung et al., 2023), and LCCAF (Kang et al., 2023) aim to reduce noise or capture stable features across predefined groups, such as sex or health status. XG-GNN (Qiu et al., 2024), GenM (Lee et al., 2023), AL-NEGAT (Chen et al., 2022), and CIA-GCL (Yu et al., 2025) target site-invariance, mitigating distributional heterogeneity across different imaging centers or sites. HSGNN achieves functional subnetwork-invariance by capturing heterogeneity in brain network connectivity and functional subdivisions (Chen et al., 2024). Although these methods have demonstrated improved robustness across groups, they all lack explicit subject-invariant modeling, preventing conquering inter-subject variability and resulting in suboptimal diagnostic performance.

**Graph Representation.** A variety of approaches have been proposed in the field of graph representation. FS2G (Mao et al., 2024), CIA-GCL (Yu et al., 2025), Contrasformer (Xu et al., 2024), and GroupBNA (Peng et al., 2024a) enhance feature extraction by leveraging federated learning, causal invariant subgraphs, contrastive augmentation, and group-aware network strategies. Another major direction relies on GT- or GNN-based encoders, including MMGDL (Cai et al., 2025), GBT (Peng et al., 2024b), ALTER (Yu et al., 2024), RGTNet (Wang et al., 2024), and CI-GNN (Zheng et al., 2024), which extract features from functional connectivity graphs at a single topological level, capturing either multi-modal information, long-range dependencies, or causally relevant subgraphs. Although these methods effectively extract features at a given level, they failed to capture subject-invariant patterns distributed across multiple topological levels.

## 3 METHOD

In this section, we present our Subject-Invariant Domain Generalization (SIDG) framework (Fig. 2), which is designed as a two-stage paradigm. We first formalize the problem definition in Sec. 3.1. Next, in Sec. 3.2, we introduce our HTE-GTR module guided by SIR loss for pre-training, which learns subject-invariant feature representation. Finally, in Sec. 3.3, we describe the fine-tuning stage, where supervised psychiatric disorder classification is performed.

### 3.1 PROBLEM DEFINITION

We define a functional connectome as a graph $\mathcal{G} = (\mathcal{V}, \mathcal{E}, \mathbf{A})$, where the node set $\mathcal{V} = \{v_1, \cdots, v_N\}$ represents $N$ regions-of-interest (ROIs), the edge set $\mathcal{E} \subseteq \mathcal{V} \times \mathcal{V}$ encodes FC relations, and the adjacency matrix $\mathbf{A} \in \mathbb{R}^{N \times N}$ stores the corresponding connectivity strengths. We consider a population of $M$ subjects $\mathcal{S} = \{s_1, \cdots, s_M\}$. For each subject $s_i$, we obtain $T$ connectome graphs, denoted by $\{\mathcal{G}_i^1, \cdots, \mathcal{G}_i^T\}$. Our goal is to learn a two-stage mapping: a pre-training encoder $f_1$ that extracts subject-invariant embeddings, followed by a fine-tuning classifier $f_2$ for psychiatric disorder prediction. Formally,

$$f_1 : \mathcal{G} \mapsto \mathbf{E} \in \mathbb{R}^{N \times d}, f_2 : \mathbf{E} \mapsto \hat{y} \in [0, 1] \tag{1}$$

### 3.2 PRE-TRAINING TOWARD SUBJECT-INVARIANT REPRESENTATION

The pre-training stage enforces subject-invariance through adopting self-supervised graph contrastive learning strategy. Specifically, the construction of positive and negative sample pairs is detailed in Sec. 3.2.1, the hierarchical topology enhanced graph transformer reconstruction (HTE-GTR) is described in Sec. 3.2.2, including (1) hierarchical graph construction, (2) level-specific

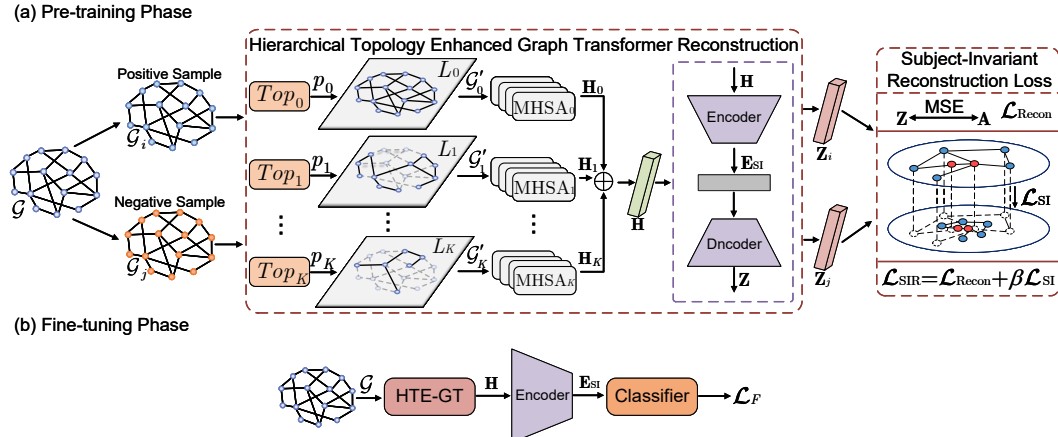

Figure 2: The overall framework of our proposed SIDG.

multi-head self-attention module, (3) hierarchical feature fusion, and (4) the encoder-decoder module, and finally, the Subject-Invariant Reconstruction (SIR) loss of pre-training stage is presented in Sec. 3.2.3.

### 3.2.1 POSITIVE AND NEGATIVE PAIRS CONSTRUCTION

Given $M$ subjects, positive pairs are constructed from graphs belonging to the same subject, while negative pairs are constructed from graphs across different subjects. Formally,

$$\mathcal{P}^+ = \{(\mathcal{G}_i^k, \mathcal{G}_i^l) \mid i \in [1, M], k \neq l\}, \mathcal{P}^- = \{(\mathcal{G}_i^k, \mathcal{G}_j^l) \mid i \neq j\}. \tag{2}$$

This construction avoids random augmentations and ensures that subject identity serves as the fundamental self-supervision signal for pre-training.

### 3.2.2 HIERARCHICAL TOPOLOGY ENHANCED GRAPH TRANSFORMER RECONSTRUCTION

**Hierarchical Graph Construction.** To avoid the introduced evidence problem caused by learnable edge weighting and the distorted connectivity problem arising from dynamic sparsification (Yuan et al., 2022), we construct hierarchical graphs via percentile-based adaptive thresholding (Ye et al., 2023; Peng et al., 2025). For each connectome $\mathcal{G} = (\mathcal{V}, \mathcal{E}, \mathbf{A})$, we generate $K$ level-specific subgraphs to capture hierarchical topology:

$$\mathcal{G}_k' = (\mathcal{V}, \mathcal{E}_k), \quad \mathcal{E}_k = \{(i, j) \in \mathcal{E} \mid A_{ij} \geq \theta_k\}, \tag{3}$$

where $\theta_k$ is the adaptive threshold determined by retaining a ratio $p_k$ of strongest edges:

$$\theta_k = \inf\{t \in \mathbb{R} \mid \sum_{i<j} \mathbb{I}(A_{ij} \geq t) \leq p_k |\mathcal{E}|\}. \tag{4}$$

Here, $t$ denotes a candidate threshold on edge weights, $\mathbb{I}(\cdot)$ is the indicator function, $|\mathcal{E}|$ is the number of edges, and $p_k$ defines the edge retention ratio at level $k$. This adaptive thresholding operation serves two purposes: (1) it removes potentially noisy weak connections that could obscure meaningful subject-invariant patterns, and (2) it generates distinct topological views.

**Level-Specific Multi-Head Self-Attention Module.** To capture subject-invariant representations while enhancing node features at each topological level, we design an $L$-layer Multi-Head Self-Attention (MHSA) module to encode each level-specific graph $\mathcal{G}_k'$. Let $\mathbf{X}_k \in \mathbb{R}^{N \times N}$ denote the input node features for graph $\mathcal{G}_k'$. The MHSA output at layer $l$ is computed as:

$$\mathbf{H}_k^l = \bigg\|_{c=1}^C \mathbf{h}_k^{l,c} \mathbf{W}_O^{l,c}, \quad \mathbf{h}_k^{l,c} = \text{Softmax}\left(\frac{(\mathbf{Z}_k^{l-1}\mathbf{W}_Q^{l,c})(\mathbf{Z}_k^{l-1}\mathbf{W}_K^{l,c})^\top}{\sqrt{d_K^{l,c}}}\right)\mathbf{Z}_k^{l-1}\mathbf{W}_V^{l,c}, \tag{5}$$

where $\mathbf{Z}_k^0 = \mathbf{X}_k$, $\|$ denotes concatenation across $C$ attention heads, $\mathbf{W}_O^{l,c}, \mathbf{W}_Q^{l,c}, \mathbf{W}_K^{l,c}, \mathbf{W}_V^{l,c}$ are learnable parameters, and $d_K^{l,c}$ is the dimension of the key vectors. The final output of the MHSA module for level $k$ is $\mathbf{H}_k = \mathbf{H}_k^L \in \mathbb{R}^{N \times N}$.

**Hierarchical Feature Fusion.** We fuse hierarchical embeddings into a unified representation as follows:

$$\mathbf{H} = \text{LayerNorm}\left(\mathbf{H}_0 + \sum_{k=1}^{K} \gamma_k \mathbf{H}_k\right), \tag{6}$$

where $\mathbf{H}_0$ encodes the original full graph. The residual connection with $\mathbf{H}_0$ ensures that information from the original graph is preserved during hierarchical processing. The coefficients $\gamma_k \in \mathbb{R}$ are learnable parameters, initialized as $\frac{1}{K}$, which adapt during training to reflect the relative importance of each level for learning subject-invariant representations. Finally, LayerNorm$(\cdot)$ normalizes the fused embeddings to stabilize training. The mathematical guarantees of hierarchical feature fusion are provided in Appendix A.1.

**Encoder-Decoder Module.** The encoder-decoder module aims to produce subject-invariant node representations while preserving the topological structure of the input graph. The encoder implemented as a graph neural network (GNN) maps the fused input $\mathbf{H}$ directly to subject-invariant embeddings:

$$\mathbf{E}_{\text{SI}} = \text{Encoder}(\mathbf{H}) \in \mathbb{R}^{N \times d}, \tag{7}$$

where $d$ is the embedding dimension. Subsequently, a GNN-based decoder reconstructs the original graph from these embeddings:

$$\mathbf{Z} = \text{Decoder}(\mathbf{E}_{\text{SI}}) \in \mathbb{R}^{N \times N}. \tag{8}$$

The overall objective of this module is twofold: (1) the encoder ensures that $\mathbf{E}_{\text{SI}}$ captures subject-invariant features in a compact latent space, and (2) the decoder enforces topological fidelity, guaranteeing that $\mathbf{E}_{\text{SI}}$ retains sufficient information to reconstruct the original graph structure.

### 3.2.3 SUBJECT-INVARIANT RECONSTRUCTION LOSS

To guide the pre-training stage, we design a subject-invariant reconstruction (SIR) loss that encourages the encoder to conquer the impact of inter-subject distributional differences while preserving other relevant graph properties. Formally, the objective is defined as

$$\mathcal{L}_{\text{SIR}} = \mathcal{L}_{\text{Recon}} + \beta \cdot \mathcal{L}_{\text{SI}}, \tag{9}$$

where $\beta$ balances the trade-off between reconstructing the original graph and enforcing subject-invariance. The convergence analysis of the SIR loss is provided in Appendix A.2.

**Reconstruction Loss.** The reconstruction loss ensures fidelity between decoder output $\mathbf{Z}$ and input adjacency $\mathbf{A}$:

$$\mathcal{L}_{\text{Recon}} = \frac{1}{N^2} \sum_{i,j=1}^{N} \|Z_{ij} - A_{ij}\|^2. \tag{10}$$

**Subject-Invariant Loss.** To explicitly remove subject-specific variations, we design a subject-invariant (SI) loss that enhances the similarity of all negative pairs while maintaining or increasing the similarity of positive pairs. Let $\mathbf{Z} = \{\mathbf{Z}_i\}_{i=1}^{B}$ denote the reconstructed adjacency matrices from the decoder for a mini-batch of $B$ graphs, where $\mathbf{Z}_i \in \mathbb{R}^{N \times N}$. For each anchor $\mathbf{Z}_i$, its positive counterpart $\mathbf{Z}_{i+}$ comes from the same subject. We first define the similarity between any two graphs using the Frobenius inner product (Montero et al., 2002) normalized by their norms:

$$s_{ij} = \frac{\langle \mathbf{Z}_i, \mathbf{Z}_j \rangle_F}{\|\mathbf{Z}_i\|_F \|\mathbf{Z}_j\|_F}, \quad \mathbf{Z}_i, \mathbf{Z}_j \in \mathbb{R}^{N \times N}. \tag{11}$$

The subject-invariant loss is formulated as

$$\mathcal{L}_{\text{SI}} = \left(\frac{1}{B} \sum_{i=1}^{B} \log\left(1 + \sum_{j \neq i} \frac{\exp(s_{ij}/\tau) - \alpha}{\exp(s_{ii+}/\tau)}\right)\right)^{-1} \tag{12}$$

where $\tau > 0$ is a temperature parameter controlling the sharpness of the similarity scaling, and $\alpha > 0$ is a corrective term introduced to stabilize the contributions from all negative pairs. The positive component aligns embeddings within each subject to preserve consistency, and the negative component pulls embeddings across subjects closer to promoting a subject-invariant representation space. The reciprocal structure further ensures numerical stability and effective gradient propagation during training.

### 3.3 FINE-TUNING FOR DOWNSTREAM CLASSIFICATION

The fine-tuning stage aims to adapt the pre-trained encoder for supervised psychiatric disorder classification. In this stage, we take the hierarchical topology enhanced graph transformer (HTE-GT) module and the Encoder from the pre-trained model. A psychiatric disorder classifier, implemented as a single-layer fully connected network, is then added on top of the encoder's output $\mathbf{h}_i$ to obtain the disorder prediction $\hat{y}_i \in [0, 1]$ for each subject. The prediction is computed via the sigmoid function:

$$\hat{y}_i = \sigma(\mathbf{D}^\top \mathbf{h}_i), \tag{13}$$

where $\mathbf{D}$ denotes the classifier weight vector and $\sigma(\cdot)$ is the sigmoid function. The model is trained by minimizing the binary cross-entropy loss over all subjects:

$$\mathcal{L}_F = -\frac{1}{M} \sum_{i=1}^{M} \big[ y_i \log \hat{y}_i + (1 - y_i) \log(1 - \hat{y}_i) \big], \tag{14}$$

where $y_i \in \{0, 1\}$ is the ground-truth label for subject $i$. During fine-tuning, all parameters are fine-tuned to ensure that the extracted features form well-separated boundaries between patients and healthy controls.

## 4 EXPERIMENTS

### 4.1 EXPERIMENTAL SETTINGS

**Datasets.** We conduct experiments on two widely used benchmark datasets, Autism Brain Imaging Data Exchange (ABIDE)[1] and Attention Deficit Hyperactivity Disorder (ADHD-200)[2]. ABIDE includes 1,009 subjects (516 ASD patients, 493 controls; age 5–64), while ADHD-200 comprises 685 subjects (243 ADHD patients, 442 controls; age 7–21). In both datasets, ROIs are defined according to the Craddock 200 atlas (Craddock et al., 2012), resulting in 200 ROIs for ABIDE and 190 ROIs for ADHD-200. To generate positive samples for graph-based learning, each ROI time series is segmented into non-overlapping sub-sequences using a 50s sliding window, inspired by prior studies showing that a window length of 30–60s yields reliable brain connectivity estimates (Preti et al., 2017). Each sub-sequence is used to build an individual brain network via pearson correlation between ROI time series, yielding multiple graph instances per subject.

**Metrics.** In this study, all methods are evaluated using a 10-fold cross-validation protocol with consistent training–testing splits. Performance is measured by five metrics: classification accuracy (ACC), sensitivity (SEN), specificity (SPE), F1 score (F1), and ROC-AUC (AUC), where higher values reflect better outcomes. Results are reported as the mean and standard deviation across 10 independent runs of 10-fold cross-validation.

**Baselines.** We compare our model with state-of-the-art (SOTA) methods: (1) graph neural network models, including BrainGB (Cui et al., 2022), CRGNN (Xia et al., 2024), CI-GNN (Zheng et al., 2024) and CIA-GCL (Yu et al., 2025); (2) graph transformer models, including BrainTrans (Kan et al., 2022), Com-BrainTF (Bannadabhavi et al., 2023), METAFormer (Mahler et al., 2023), Contrasformer (Xu et al., 2024), RGTNet (Wang et al., 2024), BrainIB (Zheng et al., 2025), GBT (Peng et al., 2024b), ALTER (Yu et al., 2024), LCCAF (Kang et al., 2023) and CAGT (Pei et al., 2025). All publicly available methods above are compared using their original code implementations.

---

[1]https://fcon_1000.projects.nitrc.org/indi/abide/
[2]https://fcon_1000.projects.nitrc.org/indi/adhd200/

**Implementation Details.** The SIDG model is optimized using the Adam optimizer with a StepLR scheduler, a learning rate of $4 \times 10^{-5}$, batch size of 128, and a maximum of 300 epochs. In the HTE-GTR module, the edge retention ratio $p = \{0.05, 0.15\}$. For the self-attention module, the number of nonlinear mapping layers and attention heads are set to 1 and 4 for ABIDE, and 1 and 2 for ADHD. The encoder–decoder architecture is configured as 200–100–200 for ABIDE and 190–100–190 for ADHD. During the pre-training stage, the learning loss incorporates a balance coefficient $\beta = 0.85$, a temperature parameter $\tau = 0.1$, and a corrective term coefficient $\alpha = 1$. All parameter configurations are determined through systematic tuning to ensure optimal performance. The experiments are implemented in PyTorch and trained on a single NVIDIA 4090 with 48 GB memory.

## 4.2 EXPERIMENTAL RESULTS

The experimental results are summarized in Tab. 1 on the two datasets. SIDG significantly outperforms all SOTA methods across all evaluation metrics. On ABIDE, it surpasses the second-best method (CIA-GCL) by 3.32%, and on ADHD-200, it exceeds the second-best method (Contrasformer) by 2.07%. These findings demonstrate the effectiveness of our two-stage learning paradigm, in which pre-training enforces subject-invariant representations and fine-tuning leverages them for accurate psychiatric disorder classification. In addition, to further assess the interpretability of our SIDG model, we provide detailed analysis in Appendix A.7.

Table 1: Performance comparison with baselines. **Bold** indicates the best results and underlining denotes the second best outcomes.

| Dataset | Method | ACC(%) | SEN(%) | SPE(%) | F1(%) | AUC(%) |
|---------|--------|--------|--------|--------|-------|--------|
| ABIDE | BrainGB | 71.07±4.92 | 72.90±6.20 | 68.73±7.36 | 70.80±4.47 | 74.93±5.10 |
| | CRGNN | 52.71±10.32 | 63.32±15.50 | 42.45±18.84 | 58.71±6.42 | 52.91±5.05 |
| | CI-GNN | 71.89±2.91 | 73.37±4.80 | 69.44±5.37 | 70.58±2.21 | 73.32±3.62 |
| | CIA-GCL | 71.95±3.36 | 76.23±8.39 | 66.73±11.92 | 71.35±3.79 | 74.26±3.88 |
| | BrainTrans | 71.90±2.77 | 75.17±8.45 | 68.33±9.58 | 75.20±2.84 | 78.80±2.59 |
| | Com-BrainTF | 70.14±4.38 | 72.83±4.15 | 67.38±4.75 | 70.01±4.49 | 71.67±6.16 |
| | METAFormer | 70.31±2.86 | 74.38±6.64 | 67.58±6.48 | 72.85±3.29 | 72.29±3.54 |
| | Contrasformer | 68.90±2.33 | 70.91±6.04 | 65.47±7.94 | 68.70±2.74 | 70.68±2.64 |
| | RGTNet | 69.52±3.51 | 70.55±4.54 | 70.00±4.52 | 71.51±2.65 | 71.05±2.45 |
| | BrainIB | 69.97±2.82 | 70.70±4.61 | 69.77±5.27 | 70.74±2.90 | 73.44±4.35 |
| | GBT | 70.06±4.96 | 73.08±7.73 | 66.24±10.05 | 72.86±2.15 | 75.80±3.79 |
| | ALTER | 70.80±4.12 | 72.68±10.24 | 68.49±9.96 | 73.61±5.52 | 78.70±2.70 |
| | LCCAF | 71.72±1.45 | 76.71±7.45 | 65.16±7.91 | 71.45±1.55 | 68.91±2.98 |
| | CAGT | 71.28±1.83 | 70.37±10.37 | 68.00±10.43 | 72.38±3.24 | 71.13±5.42 |
| | **SIDG** | **75.27±2.30** | **82.25±5.47** | **72.76±4.93** | **77.27±2.65** | **79.55±4.99** |
| ADHD-200 | BrainGB | 71.91±2.89 | 45.56±9.96 | 90.47±11.95 | 54.62±6.65 | 71.63±6.30 |
| | CRGNN | 51.65±9.78 | 62.15±16.20 | 40.78±12.98 | 56.31±5.15 | 52.78±5.56 |
| | CI-GNN | 71.03±3.05 | 53.44±10.84 | 88.69±9.76 | 66.93±4.26 | 71.95±2.97 |
| | CIA-GCL | 64.93±3.84 | 32.22±20.95 | 90.91±9.09 | 36.89±14.95 | 64.06±4.81 |
| | BrainTrans | 64.08±4.12 | 28.42±9.18 | 86.24±7.09 | 36.91±8.77 | 66.80±4.69 |
| | Com-BrainTF | 71.78±4.50 | 56.71±17.17 | 85.76±16.81 | 68.28±7.69 | 69.75±6.96 |
| | METAFormer | 70.27±2.54 | 44.09±10.01 | 91.38±12.48 | 54.66±9.45 | 69.13±4.65 |
| | Contrasformer | 72.19±2.65 | 59.53±12.73 | 87.91±13.49 | 60.26±7.32 | 72.63±3.41 |
| | RGTNet | 60.23±2.84 | 68.00±2.50 | 43.40±4.55 | 47.80±3.45 | 58.30±2.20 |
| | BrainIB | 70.07±1.56 | 56.47±8.48 | 85.32±8.14 | 58.10±4.11 | 67.28±2.72 |
| | GBT | 66.84±2.81 | 44.50±17.58 | 80.86±9.99 | 61.77±5.28 | 72.15±4.68 |
| | ALTER | 62.76±4.04 | 33.38±17.46 | 81.96±9.67 | 37.84±14.24 | 68.51±2.33 |
| | LCCAF | 57.45±4.21 | 56.12±7.42 | 65.13±7.94 | 60.40±3.25 | 55.90±3.86 |
| | CAGT | 71.65±2.34 | 73.14±4.81 | 69.83±6.63 | 71.46±2.45 | 75.46±2.94 |
| | **SIDG** | **74.26±2.68** | **73.93±3.07** | **95.55±4.84** | **73.36±2.65** | **78.86±1.46** |

## 4.3 ABLATION STUDY AND HYPERPARAMETER ANALYSIS

**Ablation Study on SIDG.** We conduct a series of ablation studies to validate the effectiveness of each component in SIDG. To assess the contribution of Hierarchical Topology Enhancement (HTE) design, we replace it with a single-level GTR, denoted as "w/o HTE-GTR". To evaluate the impact of the subject-invariant (SI) loss, we remove it while retaining only the reconstruction loss, which is necessary for reconstruction. As shown in Tab. 2, each modification leads to performance degradation, highlighting the importance of both designs. Specifically, HTE strategy effectively extracts subject-invariant patterns across multiple topological levels, aligning with the inherently hierarchical structure of brain networks. Meanwhile, the SI Loss significantly mitigates the impact of inter-subject distributional differences, improving classification accuracy.

Table 2: Ablation study of SIDG.

| Dataset | Module | ACC(%) | SEN(%) | SPE(%) | F1(%) | AUC(%) |
|---|---|---|---|---|---|---|
| ABIDE | w/o both | 68.98±3.66 | 75.75±8.98 | 61.43±10.71 | 71.03±5.36 | 68.87±8.37 |
| | w/o HTE-GTR | 71.30±5.16 | 75.88±6.46 | 66.29±7.99 | 73.00±6.02 | 69.68±5.62 |
| | w/o SI Loss | 72.35±5.25 | 77.25±7.49 | 69.10±5.41 | 74.33±7.03 | 73.59±7.85 |
| | **SIDG** | **75.27±2.30** | **82.25±5.47** | **72.76±4.93** | **77.27±2.65** | **79.55±4.99** |
| ADHD-200 | w/o both | 70.79±5.88 | 68.79±7.44 | 90.09±4.79 | 67.20±5.91 | 72.34±4.08 |
| | w/o HTE-GTR | 72.32±5.92 | 70.36±5.84 | 92.03±4.84 | 70.36±4.43 | 74.80±4.73 |
| | w/o SI Loss | 72.21±5.67 | 71.50±5.33 | 93.26±4.92 | 71.96±3.86 | 75.41±3.65 |
| | **SIDG** | **74.26±2.68** | **73.93±3.07** | **95.55±4.84** | **73.36±2.65** | **78.86±1.46** |

**Impact of Hierarchical Depth and Edge Retention.** Our experiments indicate that the model achieves optimal performance under specific hierarchical configurations (Fig. 3). In particular, the edge retention ratio $p = \{0.05, 0.15\}$ consistently yields the best results on both ABIDE and ADHD datasets (Fig. 4). The complete results of the two-level configuration are further presented in Fig. 5 and Fig. 6, with a comprehensive sensitivity analysis provided in Appendix A.3. Notably, when the hierarchical depth is less than two, performance drops significantly, indicating that a single-level topology is insufficient to capture subject-invariant representations. In contrast, increasing the depth beyond two does not lead to further gains, suggesting that excessive hierarchy introduces redundant structures rather than meaningful information. These results highlight the effectiveness of our hierarchical topology enhancement strategy, which enables HTE-GTR to fully extract subject-invariant patterns across multiple topological levels.

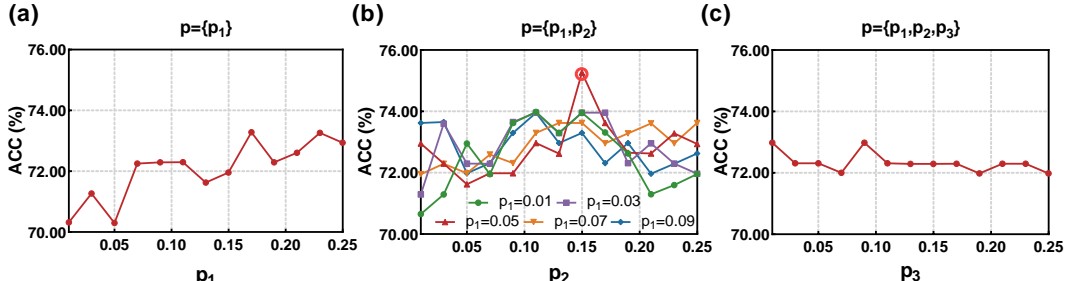

Figure 3: Impact of hierarchical depth and edge retention on ABIDE.

**Runtime and Memory Evaluation of HTE-GTR.** We performed runtime and resource profiling to assess the computational overhead of HTE-GTR (Tab. 3). Despite employing three parallel GT modules, corresponding to the original input plus the two-level configuration, the actual overhead remains moderate. Specifically, training time increases by approximately 1.25 times, inference time by 1.31 times, and memory usage by 2.46 times. These increases are substantially lower than the theoretical threefold overhead. More importantly, HTE-GTR delivers clear performance gains while

incurring only modest overhead, with ACC improving by 3.97% on ABIDE. These results underscore the necessity and effectiveness of the HTE-GTR module in enhancing model performance. In addition, we also investigated the impact of MHSA depth and the number of attention heads (Fig. 7) as well as the impact of encoder–decoder depth and dimension (Fig. 8). Due to page limitations, the detailed results are provided in Appendix A.4 and Appendix A.5.

Table 3: Comparison of runtime overhead between HTE-GTR and single-level GTR on ABIDE

| Model | Training (s/epoch) | Inference (ms/sample) | Memory (MB) | ACC (%) |
|---|---|---|---|---|
| Single-level GTR | 0.6892 | 0.6280 | 1697.66 | 71.30±5.16 |
| HTE-GTR | 0.8600 | 0.8240 | 4168.96 | 75.27±2.30 |

**Effectiveness of Subject-Invariant Modeling**  To investigate the effectiveness of the Subject-Invariant modeling, we first defined a set of quantitative metrics to measure subject variability. Following Lu et al. (2024), we computed the correlation between functional connectivity graphs across text subjects, forming an "identifiability matrix". In this matrix, the diagonal elements ($I_{self}$) capture the similarity of matched subjects, while the off-diagonal elements ($I_{other}$) capture the similarity of unmatched subjects. We further defined $dI_{self}$ as the average of the diagonal elements, $dI_{other}$ as the average of the off-diagonal elements, and $dI_{diff} = dI_{self} - dI_{other}$ to quantify the average subject identifiability. Subject Identification Accuracy (SIA) was calculated based on the ability to correctly identify subjects from these correlations. The results in Tab. 4 reveal several important insights. In the Original setting, ABIDE and ADHD exhibit high IIA, indicating substantial inter-subject variability. After encoding, both $dI_{self}$ and $dI_{other}$ increase significantly, while $dI_{diff}$ and IIA decrease markedly, demonstrating that the model effectively mitigates subject differences and extracts subject-invariant representations. Following decoding, SIA partially recovers, ensuring that the embeddings retain discriminative power necessary for downstream tasks. Collectively, these results confirm that our proposed model successfully balances the elimination of undesired subject variability with the preservation of task-relevant discriminative features, highlighting the effectiveness of the Subject-Invariant modeling for robust downstream classification. Furthermore, we conducted a comprehensive hyperparameter analysis of the SIR loss, including the effects of the balancing coefficient, the temperature parameter, and the corrective term. Detailed results are provided in Appendix A.6, with illustrative examples shown in Fig. 9 and Fig. 10.

Table 4: Effectiveness of subject-invariant modeling

| Dataset | Type | $dI_{self}$ | $dI_{other}$ | $dI_{diff}$ | SIA (ACC%) |
|---|---|---|---|---|---|
| ABIDE | Original | 0.7881 | 0.5712 | 0.2169 | 98.18 |
| | Encoder | 0.9859 | 0.9717 | 0.0141 | 79.21 |
| | Decoder | 0.8987 | 0.7891 | 0.1097 | 96.04 |
| ASD | Original | 0.5672 | 0.3248 | 0.2424 | 94.16 |
| | Encoder | 0.7102 | 0.5407 | 0.1695 | 78.59 |
| | Decoder | 0.7313 | 0.5688 | 0.1624 | 85.92 |

## 5 CONCLUSION AND FUTURE WORK

In this work, we propose SIDG, a novel framework for psychiatric disorder identification. It learns subject-invariant representations through a hierarchical topology enhanced graph transformer reconstruction module and a tailored subject-invariant reconstruction loss, effectively mitigating the impact of inter-subject distributional differences and improving generalization to target datasets. Our study provides valuable insights for brain network analysis and advances the understanding and diagnosis of psychiatric disorders. For future work, we plan to extend our framework in two directions. First, we will incorporate multi-modal brain imaging data to enhance the richness and robustness of learned representations. Second, we aim to transfer the approach to a broader range of brain network analysis tasks, thereby increasing its applicability and clinical relevance.

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

# A  APPENDIX

## A.1  MATHEMATICAL GUARANTEES FOR HIERARCHICAL FEATURE FUSION

The hierarchical feature fusion module integrates embeddings from multiple topological levels into a unified representation:

$$\mathbf{H} = \text{LayerNorm}\left(\mathbf{H}_0 + \sum_{k=1}^{K} \gamma_k \mathbf{H}_k\right), \tag{15}$$

where $\mathbf{H}_0$ is the embedding of the original full graph, $\mathbf{H}_k$ are embeddings from level-specific subgraphs, and $\gamma_k$ are learnable fusion coefficients. This fusion scheme enjoys the following four desirable mathematical properties, providing guarantees for effective subject-invariant representation learning.

**Preservation of Original Graph Information.**  The residual connection with $\mathbf{H}_0$ ensures that information from the original graph is directly retained in the fused embedding:

$$\mathbf{H} - \sum_{k=1}^{K} \gamma_k \mathbf{H}_k = \text{LayerNorm}(\mathbf{H}_0), \tag{16}$$

guaranteeing that the fusion does not discard the original full-graph representation, which is critical for downstream classification tasks.

**Adaptive Importance Weighting.**  The learnable coefficients $\gamma_k$ allow the model to assign relative importance to each hierarchical level. During training, gradient-based optimization ensures that levels contributing more to subject-invariant features are weighted higher, while less informative levels are downweighted. Formally, under standard gradient descent, the update rule

$$\gamma_k \leftarrow \gamma_k - \eta \frac{\partial \mathcal{L}}{\partial \gamma_k} \tag{17}$$

guarantees that the coefficients converge towards an optimal weighting scheme that maximizes the downstream task performance.

**Normalization Stability.**  The use of LayerNorm ensures that the fused embedding has zero mean and unit variance along each feature dimension, which mitigates covariate shift across different subjects:

$$\mathbb{E}[\mathbf{H}_{i,:}] = 0, \quad \text{Var}[\mathbf{H}_{i,:}] = 1, \quad \forall i \in [1, N], \tag{18}$$

providing numerical stability and preventing any single hierarchical level from dominating the representation due to level differences.

**Boundedness and Lipschitz Continuity.**  Given that LayerNorm and linear transformations in $\mathbf{H}_k$ are Lipschitz continuous with bounded parameters, the fusion operation $\mathbf{H}_0 + \sum_{k=1}^{K} \gamma_k \mathbf{H}_k$ is also Lipschitz continuous. Consequently, small perturbations in any level-specific embedding $\mathbf{H}_k$ induce only bounded changes in the fused representation $\mathbf{H}$:

$$\|\Delta \mathbf{H}\| \leq \left(1 + \sum_{k=1}^{K} |\gamma_k| L_k\right) \max_k \|\Delta \mathbf{H}_k\|, \tag{19}$$

where $L_k$ is the Lipschitz constant of the $k$-th level embedding module. This ensures robustness to noise or variations in individual subgraphs.

These properties guarantee that hierarchical feature fusion preserves essential graph information, adaptively balances contributions from multiple topological levels, stabilizes training through normalization, and maintains robustness. These mathematical guarantees underpin the effectiveness of HTE-GTR in learning subject-invariant representations for brain network analysis.

## A.2 CONVERGENCE ANALYSIS FOR SUBJECT-INVARIANT RECONSTRUCTION LOSS

The pre-training loss is defined as the weighted sum of reconstruction loss and subject-invariant loss:

$$\mathcal{L}_{\text{Pre}}(\theta) = \mathcal{L}_{\text{Recon}}(\theta) + \beta \, \mathcal{L}_{\text{SI}}(\theta), \tag{20}$$

where $\theta$ denotes the parameters of the HTE-GTR and encoder-decoder modules, and $\beta > 0$ is a balancing coefficient.

**Assumptions.** To rigorously analyze convergence, we make the following assumptions:

1. **Smoothness:** Both component losses are $L$-smooth with Lipschitz continuous gradients:

$$\|\nabla\mathcal{L}_{\text{Recon}}(\theta_1) - \nabla\mathcal{L}_{\text{Recon}}(\theta_2)\| \le L_1\|\theta_1 - \theta_2\|, \tag{21}$$
$$\|\nabla\mathcal{L}_{\text{SI}}(\theta_1) - \nabla\mathcal{L}_{\text{SI}}(\theta_2)\| \le L_2\|\theta_1 - \theta_2\|, \quad \forall\theta_1, \theta_2 \in \Theta, \tag{22}$$

where $\Theta$ is a compact domain of parameters ensuring numerical stability of the reciprocal-log function in $\mathcal{L}_{\text{SI}}$.

2. **Lower Boundedness:** Both losses are non-negative and bounded below:

$$\mathcal{L}_{\text{Recon}}(\theta) \ge 0, \quad \mathcal{L}_{\text{SI}}(\theta) \ge 0 \quad \Rightarrow \quad \mathcal{L}_{\text{Pre}}(\theta) \ge 0, \quad \forall\theta \in \Theta. \tag{23}$$

3. **Gradient Norm Bound:** The gradients of $\mathcal{L}_{\text{SI}}$ are finite for all $\theta \in \Theta$:

$$\|\nabla\mathcal{L}_{\text{SI}}(\theta)\| \le G_{\text{SI}} < \infty. \tag{24}$$

This is ensured by clipping or restricting $\theta$ to a domain that avoids singularities in the reciprocal term.

**Gradient Descent Update.** Let $\theta_t$ denote the parameters at iteration $t$, updated via gradient descent with learning rate $\eta > 0$:

$$\theta_{t+1} = \theta_t - \eta\nabla\mathcal{L}_{\text{Pre}}(\theta_t). \tag{25}$$

**Descent Lemma.** For $L$-smooth functions, the standard descent lemma holds (Vankov et al., 2024):

$$\mathcal{L}_{\text{Pre}}(\theta_{t+1}) \le \mathcal{L}_{\text{Pre}}(\theta_t) - \eta\|\nabla\mathcal{L}_{\text{Pre}}(\theta_t)\|^2 + \frac{\eta^2 L_{\text{total}}}{2}\|\nabla\mathcal{L}_{\text{Pre}}(\theta_t)\|^2, \tag{26}$$

where $L_{\text{total}} = L_1 + \beta L_2$ is the Lipschitz constant of the total loss.

By choosing $\eta \le \frac{1}{L_{\text{total}}}$, we obtain

$$\mathcal{L}_{\text{Pre}}(\theta_{t+1}) \le \mathcal{L}_{\text{Pre}}(\theta_t) - \frac{\eta}{2}\|\nabla\mathcal{L}_{\text{Pre}}(\theta_t)\|^2. \tag{27}$$

**Convergence Result.** Since $\mathcal{L}_{\text{Pre}}(\theta)$ is lower bounded and decreases monotonically under gradient descent, the sequence $\{\mathcal{L}_{\text{Pre}}(\theta_t)\}$ converges:

$$\lim_{t\to\infty} \mathcal{L}_{\text{Pre}}(\theta_t) = \mathcal{L}_{\text{Pre}}^* \ge 0. \tag{28}$$

Furthermore, the gradient norm vanishes in the limit:

$$\lim_{t\to\infty} \|\nabla\mathcal{L}_{\text{Pre}}(\theta_t)\| = 0, \tag{29}$$

indicating convergence to a stationary point of the pre-training loss.

**Remarks.**

- The convergence is guaranteed under the assumptions of smoothness, bounded gradients, and lower boundedness of the composite loss.
- For $\mathcal{L}_{SI}$, numerical stability is ensured by restricting the domain $\Theta$ or using gradient clipping to avoid singularities in the reciprocal term.
- In practice, adaptive optimizers such as Adam can accelerate convergence while maintaining stability.

**Interaction Analysis of Component Losses.** To rigorously understand the behavior of the composite loss, we examine the interaction between $\mathcal{L}_{Recon}$ and $\mathcal{L}_{SI}$ through their gradients. Let

$$\nabla \mathcal{L}_{Pre}(\theta) = \nabla \mathcal{L}_{Recon}(\theta) + \beta \nabla \mathcal{L}_{SI}(\theta). \tag{30}$$

Define the cosine similarity between the component gradients as

$$\cos \phi(\theta) = \frac{\langle \nabla \mathcal{L}_{Recon}(\theta), \nabla \mathcal{L}_{SI}(\theta) \rangle}{\|\nabla \mathcal{L}_{Recon}(\theta)\| \, \|\nabla \mathcal{L}_{SI}(\theta)\|}. \tag{31}$$

- If $\cos \phi(\theta) > 0$, the gradients are aligned, indicating that minimizing $\mathcal{L}_{Recon}$ also reduces $\mathcal{L}_{SI}$, resulting in a synergistic effect.
- If $\cos \phi(\theta) < 0$, the gradients are conflicting, and the weighting coefficient $\beta$ controls the trade-off. The choice of $\beta$ ensures that the update direction $\nabla \mathcal{L}_{Pre}$ still descends the total loss.

We can bound the norm of the composite gradient using the triangle inequality:

$$\|\nabla \mathcal{L}_{Pre}(\theta)\| \leq \|\nabla \mathcal{L}_{Recon}(\theta)\| + \beta \|\nabla \mathcal{L}_{SI}(\theta)\| \leq G_{Recon} + \beta G_{SI}, \tag{32}$$

where $G_{Recon}$ and $G_{SI}$ are upper bounds of the gradients. This ensures that the update step $\theta_{t+1} = \theta_t - \eta \nabla \mathcal{L}_{Pre}(\theta_t)$ remains numerically stable.

**Effective Smoothness of the SIR Loss.** Considering the interaction, the smoothness constant of the composite loss can be bounded by

$$L_{Pre} \leq L_1 + \beta L_2 + 2\beta \max_\theta \|\nabla^2 \langle \mathcal{L}_{Recon}, \mathcal{L}_{SI} \rangle\|, \tag{33}$$

where the cross-Hessian term captures second-order interactions between the losses. Under reasonable assumptions that this term is bounded, the composite loss remains $L_{Pre}$-smooth, preserving the convergence guarantees derived in the gradient descent analysis.

**Implication.** This detailed interaction analysis supports the claim that the composite loss $\mathcal{L}_{Pre}$ converges. Even if the two components exert conflicting gradients, the boundedness of each gradient and the controlled weighting $\beta$ ensure that each update step decreases the total loss, and the sequence $\{\mathcal{L}_{Pre}(\theta_t)\}$ converges to a stationary point.

A.3 OVERALL ANALYSIS OF HIERARCHICAL DEPTH AND EDGE RETENTION

We supplement the results on the impact of hierarchical layers and edge retention ratios for the ADHD dataset (Fig. 4). Consistent with the observations on ABIDE (Fig. 3), the edge retention ratio $p = \{0.05, 0.15\}$ achieves the best performance. Fig. 5 and Fig. 6 present the complete results of all two-level $(p_1, p_2)$ combinations, where each entry corresponds to the classification ACC for the respective configuration. For ABIDE, when $p_1$ is within the range from 0.01 to 0.09 and $p_2$ within 0.11 to 0.17, the performance remains stable regardless of the order of $p_1$ and $p_2$, while configurations outside this range show a noticeable decline. For ADHD, stable performance is observed over a broader range, with $p_1$ and $p_2$ from 0.11 to 0.25, as well as $p_1$ within 0.03 to 0.07 and $p_2$ within 0.13 to 0.17, again independent of the order. These results demonstrate that HTE-GTR maintains robust and stable performance across a wide spectrum of edge retention ratios. These analysis highlights the effectiveness of our hierarchical topology enhancement strategy, which enables HTE-GTR to extract subject-invariant patterns across multiple topological levels.

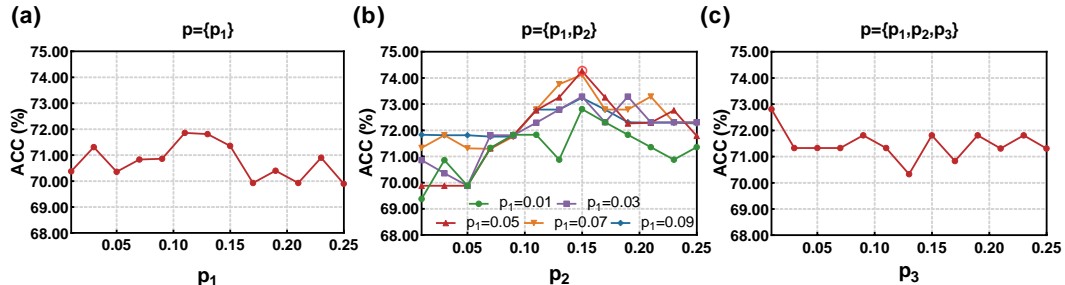

Figure 4: Impact of hierarchical depth and edge retention on ADHD.

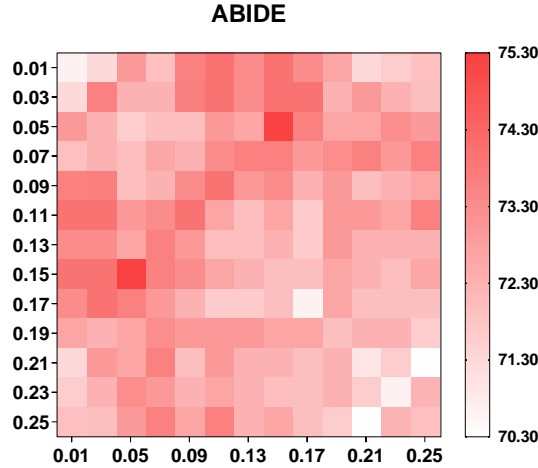

Figure 5: Complete results (ACC(%)) of two-level configuration on ABIDE.

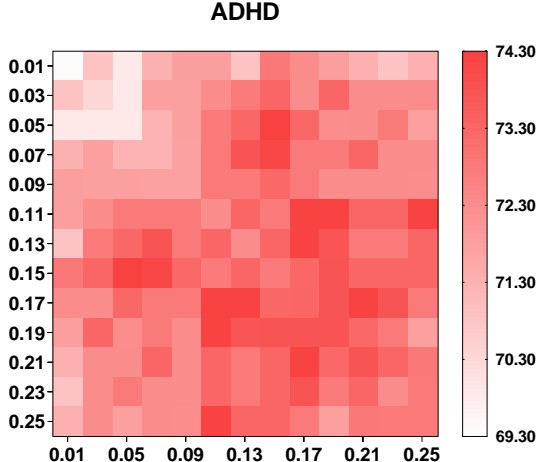

Figure 6: Complete results (ACC(%)) of two-level configuration on ADHD.

### A.4 IMPACT OF MHSA DEPTH AND NUMBER OF ATTENTION HEADS

We investigated the impact of MHSA depth and the number of attention heads on both datasets (Fig. 7). Regarding MHSA depth, we found that shallow models consistently outperform deeper architectures across both datasets, indicating that for the given node sizes (ABIDE: 200, ADHD: 190), shallow models are more suitable. Deeper architectures may lead to overfitting in this setting. However, they could offer advantages in scenarios with higher-resolution brain networks, where the

representational capacity of a shallow model may become insufficient. Concerning the number of attention heads, the choices are constrained by factors of the respective node counts (ABIDE: 200, ADHD: 190). Therefore, we evaluated 1, 2, 4, and 5 heads for ABIDE, and 1, 2, and 5 heads for ADHD. Optimal performance was achieved with 4 heads on ABIDE and 2 heads on ADHD, suggesting that excessive heads may introduce redundancy and computational overhead without further gains.

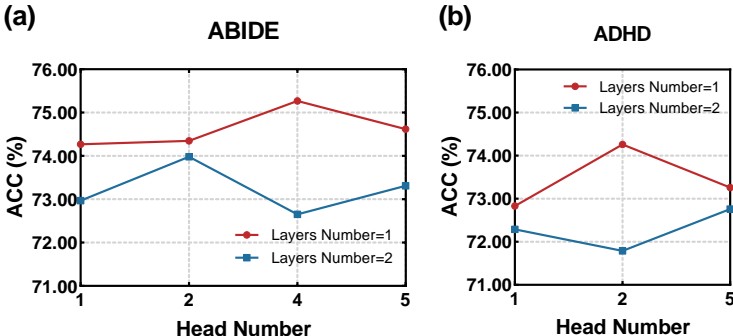

Figure 7: Impact of MHSA depth and number of attention heads.

## A.5 IMPACT OF ENCODER–DECODER DEPTH AND DIMENSION

We investigated the impact of Encoder–Decoder depth and dimension on classification performance. As shown in Tab. 5 and Tab. 6, increasing the depth from a single layer ($200 \rightarrow 100$) to two layers ($200 \rightarrow 100 \rightarrow 64$) leads to a noticeable drop in ACC on both datasets, indicating that deeper architectures may introduce overfitting for the current node sizes. Regarding the hidden dimension, we studied a single-layer Encoder–Decoder with dimensions ranging from 65 to 125 (Fig. 8). The ACC varied non-monotonically with dimension, achieving optimal performance at 100 for both ABIDE and ADHD. This finding suggests that a moderate hidden dimension strikes a balance between sufficient representational capacity and avoiding overfitting, providing robust performance across datasets.

Table 5: Impact (ACC(%)) of Encoder–Decoder depth on ABIDE

| Architecture | ABIDE |
|---|---|
| $200 \rightarrow 100$ | $75.27 \pm 2.30$ |
| $200 \rightarrow 100 \rightarrow 64$ | $70.65 \pm 5.73$ |

Table 6: Impact (ACC(%)) of Encoder–Decoder depth on ADHD

| Architecture | ADHD |
|---|---|
| $190 \rightarrow 100$ | $74.26 \pm 2.68$ |
| $190 \rightarrow 100 \rightarrow 64$ | $71.81 \pm 6.48$ |

## A.6 HYPERPARAMETER ANALYSIS OF COMPOSITE LOSS IN PRE-TRAINING

We conducted a comprehensive hyperparameter analysis of the SIR loss on ABIDE (Fig. 9) and ADHD (Fig. 10), examining the effects of the balancing coefficient $\beta$, the corrective term $\alpha$, and the temperature parameter $\tau$. For the balancing coefficient $\beta$, optimal performance is achieved at 0.85 for both ABIDE and ADHD. Specifically, ABIDE maintains stable performance when $\beta$ ranges from 0.8 to 0.95, while ADHD remains stable when $\beta$ is between 0.8 and 0.9. Beyond these ranges, performance significantly declines, demonstrating the model's sensitivity to the trade-off

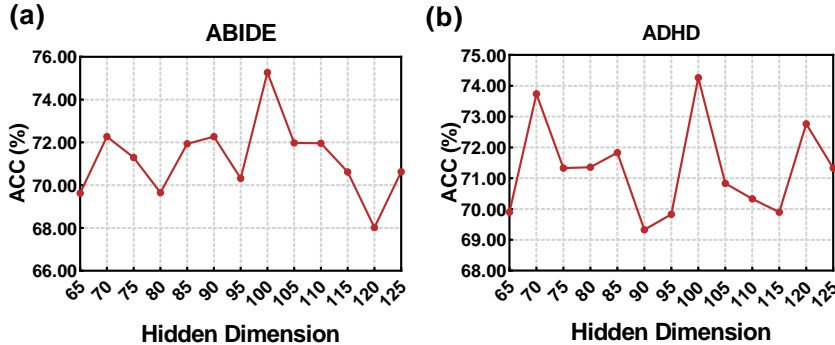

Figure 8: Effect of Encoder–Decoder hidden dimension in a single layer.

between reconstructing the original graph and enforcing subject-invariance. For the corrective term $\alpha$, the optimal value is 1 for both datasets. ABIDE remains stable when $\alpha$ is between 1.0 and 1.2, and ADHD remains stable when $\alpha$ is between 0.8 and 1.4, with significant drops observed outside these ranges. This indicates that $\alpha$, introduced to stabilize contributions from all negative pairs, effectively ensures robust learning when appropriately tuned. For the temperature parameter $\tau$, the model achieves the best performance at 0.1 for both ABIDE and ADHD, with notable performance degradation observed for values smaller or larger than 0.1. This highlights the importance of $\tau$ in controlling the sharpness of the similarity scaling and maintaining stable subject-invariant feature learning.

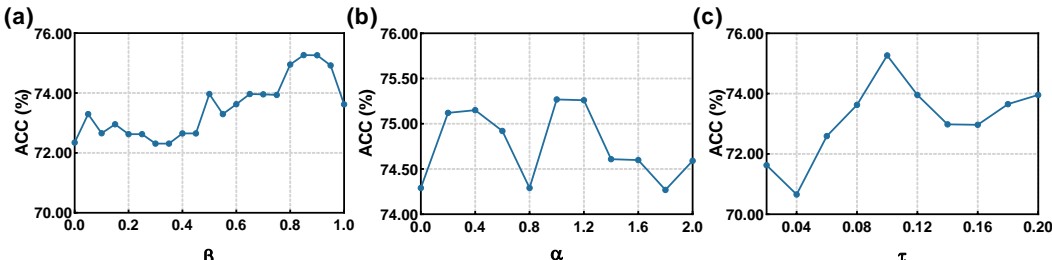

Figure 9: Hyperparameter analysis of SIR loss on ABIDE.

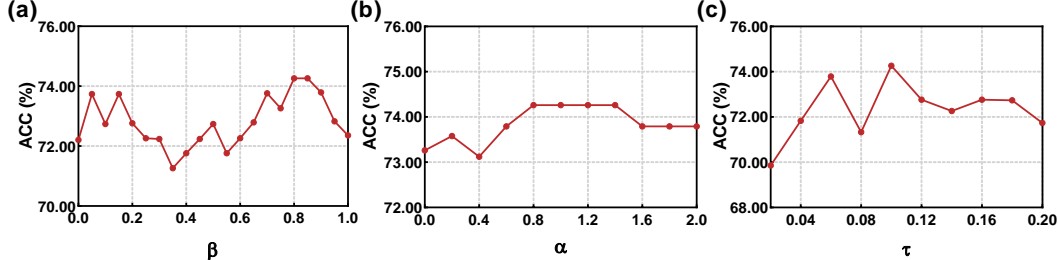

Figure 10: Hyperparameter analysis of SIR loss on ADHD.

## A.7 INTERPRETABILITY ANALYSIS OF SIDG MODEL

Due to the lack of atlas-based labels in ABIDE, our interpretability analysis focuses on ADHD-200. Specifically, we extracted the top 10 brain regions with the highest learned attention scores, and following the functional modules defined in (Dosenbach et al., 2010), the ROIs are classified into six modules, including visual cortex (Vis), motion control (MC), cognitive control (CC), auditory cortex (Aud), language processing (LP), and executive control (EC). The attention scores are primarily

concentrated in the MC and EC subnetworks (Tab. 7). This pattern aligns well with the neuropathological basis of attention deficit hyperactivity disorder, as MC regions are linked to hyperactivity symptoms (Rubia et al., 1999), while EC regions are central to deficits in attention regulation and cognitive control (Francx et al., 2015). The fact that SIDG autonomously focuses on these biologically meaningful regions demonstrates its interpretability, since it does not rely on arbitrary patterns but emphasizes known neurocognitive substrates that are closely related to disease diagnosis.

Table 7: Top 10 important ROIs of ADHD. "No." represents the descending sorting order, "Label" is the default order of ROI in the atlas.

| No. | Label | Network | ROI |
| --- | --- | --- | --- |
| 1 | 35 | MC | lat_cerebellum_128 |
| 2 | 41 | MC | med_cerebellum_138 |
| 3 | 1 | Vis | occipital_146 |
| 4 | 99 | EC | vlPFC_12 |
| 5 | 29 | MC | inf_cerebellum_151 |
| 6 | 100 | EC | dlPFC_16 |
| 7 | 90 | EC | dACC_27 |
| 8 | 49 | MC | basal_ganglia_39 |
| 9 | 33 | MC | inf_cerebellum_121 |
| 10 | 27 | MC | inf_cerebellum_155 |

## A.8 THE USE OF LARGE LANGUAGE MODELS (LLMS)

In this work, we utilized Large Language Model to assist in polishing the manuscript. All scientific content and interpretations were independently authored by the research team.

## A.9 POSSIBLE NEGATIVE SOCIAL IMPACTS

As the research in this paper focuses on the diagnosis of Autism Spectrum Disorder (ASD) and Attention Deficit Hyperactivity Disorder (ADHD), it is important to consider the potential negative social impacts of this work, even though the current research remains at a scientific stage and has not been applied in practice. One major concern is incorrect diagnosis. AI-based methods are inherently prone to errors, which cannot be entirely eliminated. An erroneous diagnosis could have serious consequences for individuals and society. Therefore, AI tools should be used solely as diagnostic aids rather than decision-makers, with final clinical judgments always made by qualified medical professionals.

Another concern involves the leakage of private information. Datasets related to ASD and ADHD often contain highly sensitive personal information. Unauthorized disclosure of such data could lead to unpredictable and significant impacts on both individuals and society. To mitigate this risk, all identifying information of subjects in this study has been completely anonymized and is not accessible to the research team, ensuring that privacy is strictly protected.

