# OpenReview forum: "Subject-Invariant Domain Generalization for Psychiatric Disorder Identification"
_ICLR.cc/2026/Conference — ICLR 2026 Conference Withdrawn Submission_

### Official Review · Reviewer_rdxb · 2025-10-26

**Soundness:** 2
**Presentation:** 2
**Contribution:** 3
**Rating:** 4
**Confidence:** 4

**Summary:**

This paper proposes a novel Subject-Invariant Domain Generalization (SIDG) framework for psychiatric disorder identification using functional brain networks. The method addresses the critical challenge of inter-subject distributional differences in brain network data by introducing a two-stage learning paradigm that combines hierarchical graph representation with subject-invariant representation learning. The approach demonstrates strong performance on public ABIDE and ADHD datasets, outperforming state-of-the-art methods.

**Strengths:**

1. The HTE-GTR model effectively captures hierarchical information in brain networks through multi-level topological graph construction and graph attention mechanisms, providing an innovative approach to brain network analysis.

2. The proposed SIR loss function, combining reconstruction loss and subject-invariant loss, offers a robust training objective that strengthens cross-subject representation convergence and demonstrates good generalization ability.

3. The experimental design is comprehensive, including numerous comparison experiments, ablation studies, and detailed hyperparameter sensitivity analysis, showcasing the method's robustness and adaptability.

4. The implementation is open-sourced and evaluated on publicly available benchmark datasets, facilitating reproducibility and fair comparison.

**Weaknesses:**

1. The construction of positive-negative pairs aims to maintain subject-identity consistency while suppressing within-subject fluctuations, but this strategy is not equivalent to true "de-subjectization." The manuscript lacks statistical evidence sufficient to guarantee de-subjectization.

2. Percentile-based adaptive thresholding without stability selection or sparsity priors may treat diagnostically informative weak edges as noise, potentially removing medium-to-low strength pathways relevant to psychiatric disorders.

3. The claimed subject invariance is vulnerable to acquisition-site confounding and batch effects. The same-subject positive pairing, reconstruction of raw connectomes, and percentile-based thresholding can preserve scanner/protocol signatures, potentially inducing embeddings that align with domain (site) rather than pathology.

4. The SIR loss primarily aggregates established objectives (MSE, Frobenius inner product, and InfoNCE), showing practical utility but offering limited methodological novelty.

5. Equation (12) defines L_SI as a variant of InfoNCE, but in its current form, the gradient descent direction conflicts with the stated goal of "maintaining or increasing the similarity of positive pairs." The motivation and functional role of α are not clearly explained.

6. The mathematical derivation from Eq. (15) to Eq. (16) is invalid due to the nonlinear nature of LayerNorm. A more precise formulation is needed.

7. The introduction lacks concise explanations of key concepts such as "domain generalization" and "domain-invariant representations," which may hinder accessibility.

8. The term "subject-invariant representations" is emphasized throughout but lacks an explicit definition in early sections, affecting clarity.

9. The statement "most current methods focus on a single topological level" is somewhat overstated, as some existing works have explored hierarchical topological modeling.

10. The motivation for constructing positive and negative pairs is not clearly explained, and the logical flow connecting components like positive/negative pair construction and sliding-window segmentation to the model pipeline needs improvement.

11. The implementation details section lacks justification for hyperparameter selections, and performance sensitivity to these parameters is not adequately discussed.

12. The experimental comparison lacks classical baseline methods, and evaluation is limited to only two public datasets without generalization experiments on unseen datasets.

13. The framework diagram (Figure 2) should arrange the encoder-decoder module horizontally rather than vertically for better readability.

14. The presentation of experimental results is relatively homogeneous; incorporating qualitative visualizations or more diverse evaluation metrics would be beneficial.

15. In Section 3.1, the symbols E and d in formula (1) are ambiguous, reducing readability.

16. Section 3.2.1 lacks detailed explanation of positive and negative sample construction and its significance, making model inputs unclear and reproduction difficult.

17. The Effectiveness of Subject-Invariant Modeling section lacks specific explanation of how SIA is calculated and contains incorrect notation ('IIA' should be 'SIA').

18. The data preprocessing process lacks specific explanation, particularly regarding how multiple graph instances per subject are used in model training.

19. Some charts (e.g., Figure 3) have poor readability, and the presentation format of experimental results is relatively simple.

**Questions:**

1. The manuscript employs positive-negative pair training and a loss function rooted in contrastive learning, yet neither the main text nor the related-work section explicitly acknowledges or discusses this basis. Could you clarify why this is not addressed?

2. In the manuscript, the three-level (p1, p2, p3) experiment appears incomplete in either its conduct or its presentation; could you clarify the reason?

3. The framework diagram does not specify the input scheduling of positive-negative pairs: are they fed into the network simultaneously (in parallel) or sequentially (in turn)? Please clarify.

4. In constructing the brain network, why was Pearson correlation chosen rather than alternative network-construction methods? Please clarify the rationale and its implications for the results.

5. Are positive and negative sample pairs inputted into the model simultaneously or separately while training? Why divide samples into positive and negative samples? What is the specific model training process like?

6. How is the subject-invariant loss obtained and why is it set in the form of formula (12)?

7. In the fine-tuning stage of the model, will all parameters be adjusted or will a portion of the parameters be fixed and only the other part be adjusted?

8. How is the Subject Identification Accuracy (SIA) specifically calculated? Please provide the detailed calculation method.

9. What is the specific process for using multiple graph instances per subject in model training? Are they used independently or aggregated in some way?

---

### Official Review · Reviewer_uBnz · 2025-10-29

**Soundness:** 3
**Presentation:** 3
**Contribution:** 2
**Rating:** 4
**Confidence:** 3

**Summary:**

The author proposed a two-stage Subject-Invariant Domain Generalization (SIDG) model with two-stage training. In the first stage, the model learns subject-invariant information with pre-training with the Hierarchical Topology Enhanced Graph Transformer Reconstruction (HTE-GTR) module, along with Subject-Invariant Reconstruction (SIR) loss. Then, in the second stage, a finetuning of the pre-trained model is performed to adapt the generalized model to dataset-specific information.

**Strengths:**

- In the related works section, a comprehensive comparison against group invariance is done.

- The overall method is clear and well-motivated. Figure 1 is a nice example of how the method may be useful in real-world cases.

- Ablations of various hyperparameter choices are done in App. A.

**Weaknesses:**

- The main issue is that I still struggle to see the exact novelty of subject-invariant training. First of all, the method is not precisely subject-invariant, as the representation is not exactly the same across views of the same patient. It’s just learned via contrastive learning. In most literature, -invariance usually means the representations across views are either exactly the same or guaranteed by a theoretical lower bound. I see some empirical evaluations are done in sec. 4.3, which is good. Sec. A.3 includes a comprehensive, but confusing proof. In particular, the proof is so general that it is hard to see how this is different from proving “ML models can learn from multiple losses. It would be nice if the authors can elaborate more on the proof, particularly expanding on how this applies to the authors’ specific method.

- Extending from 1, it is potentially misleading to claim the method as subject-invariant. The author needs a better comparison of the method against various graph contrastive learning methods, such as the various methods outlined in this benchmark paper https://arxiv.org/abs/2109.01116.

- I like the pre-training -> finetuning paradigm. It would be nice if the pretraining-only model could be better analyzed. Right now, it is mainly evaluated via downstream tasks.

**Questions:**

See weaknesses. I would consider raising my score if the authors can explain clearly how the methods distinguishes itself from other multi-view graph contrastive learning methods.

---

### Official Review · Reviewer_Q98H · 2025-10-31

**Soundness:** 2
**Presentation:** 3
**Contribution:** 2
**Rating:** 4
**Confidence:** 4

**Summary:**

This paper introduces a Subject-Invariant Domain Generalization (SIDG) framework for identifying psychiatric disorders like ASD and ADHD using functional brain networks from fMRI data. It employs a pre-training stage with a Hierarchical Topology Enhanced Graph Transformer Reconstruction (HTE-GTR) module to learn subject-invariant representations across hierarchical topological levels, guided by a Subject-Invariant Reconstruction (SIR) loss. This is followed by fine-tuning for classification. Experiments on ABIDE and ADHD-200 claim consistent improvements over GNN/Transformer baselines, plus ablations and limited runtime profiling.

**Strengths:**

1. Novel Perspective on Invariance. The focus on subject-invariant representations differentiates it from prior work on group-invariant (e.g., sex or health-status) or site-invariant models, providing a fresh angle to address inter-subject variability in brain network data.

2. Up-to-Date Related Work. The citations are comprehensive and include recent publications (up to 2025), reflecting a strong grasp of the current literature in graph-based learning for neuroimaging.

3. Relevant Baselines. Comparisons are made against state-of-the-art methods from 2023–2025, such as CIA-GCL, BrainIB, and Contrasformer, ensuring the evaluation is contemporary.

**Weaknesses:**

1. Limited Technical Novelty. The core contribution, the Subject-Invariant (SI) Loss, feels incremental, as it inverts typical contrastive objectives in a way that echoes existing self-supervised graph learning techniques. Overall, the framework combines familiar elements (e.g., hierarchical graphs, MHSA, reconstruction losses) without groundbreaking innovations.

2. Risk of Representation Collapse in SI Loss. The SI loss's negative term explicitly pulls embeddings across subjects closer to enforce invariance, which could reward representation collapse, potentially mapping all subjects to a low-variance subspace and losing discriminative information that meaningfully differs between classes (e.g., patients vs. controls). Although the reconstruction term and theoretical convergence analysis (in Appendix A.2) attempt to counteract this, the paper lacks rigorous empirical validation, such as checks via nearest-neighbor diversity, feature variance, or centered kernel alignment, to confirm that discriminative power is preserved.

3. Poor Interpretability and Attribution. It is unclear how much the performance gains stem from the subject-invariant aspect. Ablation studies suggest that removing the Subject-Invariant Loss results in only marginal improvements over baselines, raising doubts about its impact. Critically, there are no discovered biomarkers, visualizations of invariant features, or analyses to interpret what the model learns, limiting clinical relevance. Additionally, only two datasets (ABIDE and ADHD-200) are used, restricting generalizability.

4. Mismatch in Domain Generalization Claims. The paper positions SIDG as a DG method to handle inter-subject/site distribution shifts, but evaluations rely solely on in-dataset 10-fold CV. There is no demonstration of out-of-domain robustness, such as leave-one-site-out (LOSO) cross-validation or cross-dataset transfer (e.g., training on ABIDE and testing on ADHD-200), which is essential for validating DG approaches.

**Questions:**

1. Could the authors provide more detailed ablations to isolate the contribution of the Subject-Invariant Loss, perhaps comparing it directly to standard contrastive losses like NT-Xent?

2. Why no visualizations or biomarker discovery? For instance, how do the learned representations differ across subjects, and what brain regions show invariant patterns?

3. Given the DG framing, why was evaluation limited to in-dataset CV? Have you tested LOSO or cross-dataset scenarios to show true generalization?

4. Could you provide empirical evidence that the SI loss does not lead to representation collapse? How do you ensure that pulling negative pairs closer doesn’t undermine class discriminability?

---

### Official Review · Reviewer_nGVh · 2025-10-31

**Soundness:** 2
**Presentation:** 2
**Contribution:** 2
**Rating:** 4
**Confidence:** 3

**Summary:**

This paper proposed a two-stage SIDG framework. This framework includes two modules: HTE-GTR and SIR modules. Specifically, the HTE-GTR enables to learn the subject-invariant representations. In addition, it also learns features from multiple topological levels to obtain more comprehensive and accurate representations. The SIR loss constrains the model to maintain consistent subject representations while enhancing its discriminative ability for downstream tasks. Finally, comparative and ablation experiments validate the effectiveness of the proposed method.

**Strengths:**

1.The proposed method is well-motivated and technically sound.

2.The proposed method is supported by clear mathematical proofs and theoretical analysis.

Handling the distribution discrepancy between source and target data is an important problem

**Weaknesses:**

1.Would adopting such a two-stage design lead to an increase in computational overhead? I am concerned that this may require substantial computational resources, potentially making the proposed method less efficient. It would be helpful if the authors could provide additional experiments or analyses to demonstrate the efficiency of the proposed method—for example, by comparing the training time with other baseline methods.

2.In the experimental setup section, the paper mentions “a maximum of 300 epochs.” It is not entirely clear whether this means that different datasets use different numbers of epochs (with 300 being the upper limit), or that an early-stopping strategy is applied with 300 epochs as the maximum training limit. Moreover, it would be helpful to clarify whether this refers to the pre-training or fine-tuning stage. Providing more details on this part would improve the reproducibility of the proposed method.

3.The manuscript contains several typographical and formatting issues. For example, the symbol E appears to be undefined. A colon was omitted before the equation.

**Questions:**

See above.

---

### Note · Authors · 2025-11-17

I have read and agree with the venue's withdrawal policy on behalf of myself and my co-authors.